# Activity-Based Probes for Proteases Pave the Way to Theranostic Applications

**DOI:** 10.3390/pharmaceutics14050977

**Published:** 2022-04-30

**Authors:** Georgia Sotiropoulou, Eleni Zingkou, Evangelos Bisyris, Georgios Pampalakis

**Affiliations:** 1Department of Pharmacy, School of Health Sciences, University of Patras, 26500 Rion-Patras, Greece; zingkou@upatras.gr (E.Z.); e.bissyris@gmail.com (E.B.); 2Department of Pharmacognosy-Pharmacology, School of Pharmacy, Aristotle University of Thessaloniki, 54124 Thessaloniki, Greece

**Keywords:** activity-based probes (ABPs), kallikrein-related peptidases (KLKs), cathepsins, phosphonates, proteases, activography, theranostics

## Abstract

Proteases are important enzymes in health and disease. Their activities are regulated at multiple levels. In fact, proteases are synthesized as inactive proenzymes (zymogens) that are activated by proteolytic removal of their pro-peptide sequence and can remain active or their activity can be attenuated by complex formation with specific endogenous inhibitors or by limited proteolysis or degradation. Consequently, quite often, only a fraction of the protease molecules is in the active/functional form, thus, the abundance of a protease is not always linearly proportional to the (patho)physiological function(s). Therefore, assays to determine the active forms of proteases are needed, not only in research but also in molecular diagnosis and therapy. Activity-based probes (ABPs) are chemical entities that bind covalently to the active enzyme/protease. ABPs carry a detection tag to enable localization and quantification of specific enzymatic/proteolytic activities with applications in molecular imaging and diagnosis. Moreover, ABPs act as suicide inhibitors of proteases, which can be exploited for delineation of the functional role(s) of a given protease in (patho) biological context and as potential therapeutics. In this sense, ABPs represent new theranostic agents. We outline recent developments pertaining to ABPs for proteases with potential therapeutic applications, with the aim to highlight their importance in theranostics.

## 1. Proteases and Degradome

Proteases are enzymes that catalyze the hydrolysis of peptide bonds. They are involved in almost all biological processes in an organism. The term “degradome” was introduced to describe “the complete set of proteases that are expressed at a specific time by a cell, tissue or organism” [1]. The human genome encodes 553 proteases that are classified based on the catalytic mechanism into the following five categories: aspartic, metallo, cysteine, serine, and threonine proteases. A wide spectrum of pathologies is characterized or triggered by abnormal expression and/or activation of proteases and/or endogenous inhibitors of proteases, thus, tight regulation of proteolysis is indispensable for maintenance of homeostasis in biological systems [2].

Proteases are regulated at multiple levels that include: (i) the transcriptional, (ii) the post-transcriptional (miRNAs, alternative polyadenylation, etc.), and (iii) the post-translational level, i.e., proteases are synthesized as pro-enzymes that require activation. The activity of mature enzymes is further regulated by complexation with endogenous inhibitors, degradation, pH changes, and chemical modifications (e.g., phosphorylation) that affect their activity [3] (Figure 1).

In this sense, an assay that would allow determination of the amount of the active protease could enhance the efficiency of diagnosis and treatment, since it is the fraction of the active protease that is relevant to biological function(s). Contrary to ABPs, antibody-based assays detect the total content of the protease in a biological/clinical sample, independent on activation status.

## 2. Activity-Based Probes

Activity-based probes (ABPs) are small organic molecules that bind to the active form of a target enzyme by covalent bonding. They are composed of three parts: (i) an electrophilic reactive group or warhead, (ii) a spacer that may incorporate a recognition sequence to selectively bind to a specific target enzyme that can be a protease, and (iii) a detection tag. Tags are classified as “fluorescent” (e.g., fluorescein) or “affinity handle” (e.g., biotin) [4]. Various warheads used to target proteases are shown in Figure 2. Phosphonates are the most widely used warheads for serine proteases, while acyloxymethyl ketones are mainly used to target cysteine proteases [4]. The aforementioned proteases cleave their corresponding substrates via a catalytic mechanism that involves the formation of covalent bonds between the catalytic residue (serine or cysteine, respectively) of the active site and the substrate. It must be noted that ABPs have been developed for the metalloproteinase 12 (MMP12) that lacks a canonical nucleophile in its active site but contains Zn^2+^ instead. To design the ABP, a known competitive inhibitor of MMP12 was modified to encompass a cleavage linker conjugated to a fluorescent tag (i.e., cyanine 3). Once the ABP is bound onto the active site, cleavage of the linker occurs, and the released cyanine 3-carrying group covalently modifies neighboring nucleophiles (side chains of various residues that face towards the active site), a process known as proximity driven reaction [5]. Finally, there are probes for proteases known as substrate probes. These molecules are fluorescently quenched peptide substrates encompassing the protease substrate recognition sequence. Cleavage of the peptide by the active protease releases the quencher and the proteolytic activity is quantified by monitoring fluorescence emission [6].

Originally, alkylphosphonofluoridates were developed as ABPs for serine hydrolases [7]. These first generation ABPs bind rapidly to the recognized enzyme but lack specificity. An example of such ABPs is the B24P that was used to detect the secretion of active KLK6 in primary neuronal cortical cells infected with a genetically engineered adenovirus driving the expression of preproKLK6 [8]. Second generation *O,O*-diphenyl phosphonate ABPs showed specificity for certain classes of serine proteases. These molecules carry a phosphonate analogue of the amino acid that is preferred in the P1 position. In some cases, a phosphonylated dipeptide (encompassing the P2-P1 positions) may also be used, as in the case of biotin-Pro-Lys-P(O)(OPh)_2_ (also referred to as Bio-PK) [9]. To achieve specificity for a single protease, third generation peptidyl phosphonate ABPs were generated that encompass a longer recognition sequence (usually tetrapeptides). The chemical characteristics of first, second, and third generation ABPs are shown in Figure 3.

Attempts to shorten the reaction times between the ABP and the targeted protease focused on the replacement of one *O*-diphenyl group by a fluorine atom that is a better leaving group. Nevertheless, the generated phosphonofluoridate peptidyl analogues are characterized by low stability in aqueous solutions and rapid decomposition [10]. In conclusion, to design a protease-specific ABP it is necessary to define the peptide substrate specificity of the protease. The approaches used to map cleavage sites recognized by different proteases and their corresponding specificities are depicted in Figure 4. Here, it should be mentioned that positional scanning libraries containing both natural and unnatural amino acids have been used to develop an ABP for the main protease (M^pro^) of SARS-CoV-2 that is one of the most important targets for the development of antiviral drugs. Initially, screening of the library identified the best substrate that was converted to an ABP by addition of biotin or fluorescent (cyanine-5 or bodipy) tags at the *N*-terminus and a vinyl sulfone warhead at the *C*-terminus [11].

ABPs for proteases can be applied for analytic purposes in Western blotting, SDS-PAGE coupled to fluorescence imaging, confocal laser scanning microscopy, and activity-based proteomic profiling as, for example, described for ABPs targeting SUMO proteases [12]. The ABP-enzyme adduct can also be detected by monitoring fluorescence emission following size exclusion chromatographic separation of the reaction products between the ABP and the protease, as demonstrated for the phosphonate ABP specific for matriptase [13].

**Figure 4 pharmaceutics-14-00977-f004:**
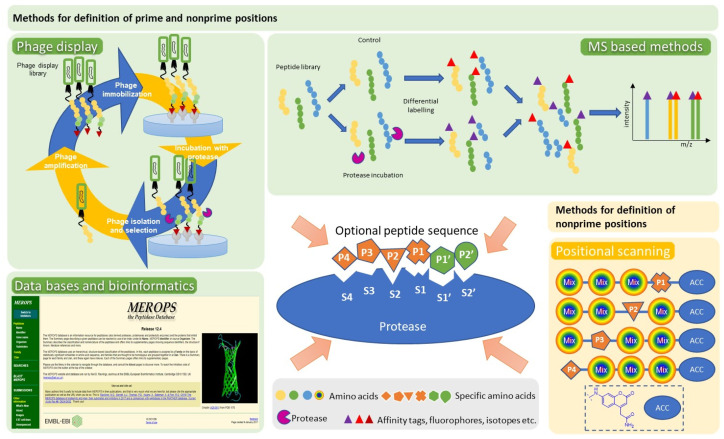
Methods used to map the substrate specificity of a given protease. The most frequently applied methods are shown. Phage display [14], mass spectrometry-based approaches [15], and mining of the MEROPS database [16,17] have been proven valuable for identification of the prime and non-prime positions. On the other hand, positional scanning of synthetic combinatorial libraries (PS-SCL) [18] is applied to identify the non-prime positions. Non-natural amino acids can be incorporated in PS-SCL libraries.

## 3. Click Chemistry-Based ABPs

In this approach, the ABP is comprised of two parts, one that contains the reactive group including the spacer and the recognition sequence, if required, and the other is the tag. The reactive group reacts with the target protease, then, the tag forms the adduct for enzyme labeling. Biorthogonal click chemistry based on the 1-alkyne and azide moieties is applied to promote the reaction between the tag and the warhead-protease adduct. The principle of click chemistry-based labeling of active serine hydrolases is depicted in Figure 5. This assay was tested with active kallikrein-related peptidase 7 (KLK7) [19]. The advantage of click chemistry-based ABPs lies in the fact that molecules are smaller in size and exhibit enhanced cell permeability, therefore, more suitable for in vivo applications, such as in vivo imaging. For preclinical in vivo imaging, copper-free click chemistry will offer a great potential, since it does not require the addition of catalysts. Copper-free click chemistry takes advantage of the smallest stable cycloalkynes, the cyclooctynes. The reaction between the azide and the cyclooctyne is biocompatible, and simpler than the reaction between the azide and the 1-alkyne, because the [3+2] cycloaddition occurs in the absence of catalysts [20].

## 4. ABPs as Theranostics: Conceptual Framework

ABPs are suicide inhibitors of proteases. Nowadays, suicide inhibitor-based drugs became increasingly popular [21]. ABPs could be exploited dually for therapeutic and diagnostic purposes and represent new theranostic (*the*rapy + diag*nostic*) agents. The therapeutic potential of ABPs has been demonstrated for the KLK and the cathepsin protease families, as elaborated in the following section. It should be noted that the first report on the utility of ABPs for functional studies appeared almost 30 years ago [16] but it was not pursued further. At that time, the authors synthesized the ABP biotin-Aca-Aca-Phe-Leu-PheP(OPh)_2_ (where Aca represents the 6-aminocaproic acid) that selectively inhibited the chymase. In a functional assay, inhibition of chymase by this ABP diminished the lysis of erythrocytes mediated by granules [22].

### 4.1. Kallikrein-Related Peptidases

Kallikrein-related peptidases (KLKs) constitute the largest family of serine proteases in humans, encompassing 15 enzymes with either trypsin- or chymotrypsin-like activity. KLK3 or prostate-specific antigen (PSA) is a well-known KLK, and a circulating biomarker measured routinely for diagnosis of prostate cancer and monitoring therapeutic response. The expression of KLKs is regulated at multiple levels. Specifically, at the level of transcription several KLKs are controlled by epigenetic mechanisms [23,24], utilization of alternative promoters [25], and/or alternative splicing that yields multiple transcripts from a single gene sequence [25,26]. KLK proteins are synthesized as inactive zymogens which are activated by other active KLKs or other proteases individually or/and in proteolytic cascade pathways [27,28]. The involvement of KLK proteases in various (patho)physiologies, including seminal plasma liquefaction, epidermal (over)desquamation and inflammation, neurodegeneration, and different types of cancer is well-established [29].

Despite their important roles in (patho)physiology, attempts to generate ABPs targeting specific KLKs, that could be used to determine the active forms of these enzymes, were described only very recently. In fact, the first ABP to target KLKs was the compound DKFZ-633 that is specific for KLK6 and was based on a depsipeptide inhibitor of KLK6 that was modified to carry a 1-alkyne moiety, thus, detection was achieved with a click chemistry-based approach [30]. Then, a KLK13 specific ABP was developed (Figure 6), which contained the KLK13 specific tetrapeptide (Val-Arg-Phe-Arg) substrate modified with a chloromethyl ketone warhead and a biotin tag [31], and a KLK14 specific phosphonate tetrapeptide ABP [32]. Nonetheless, in all these cases, the ABPs were not used for therapeutic purposes.

Recently, two phosphonate peptidyl ABPs for KLK7 were generated by our group based on the Phe-Phe motif and a biotin tag [16,33]. In a mouse model of Netherton syndrome (NS) it was demonstrated that this KLK7 ABP could elicit a therapeutic effect in vivo by inhibition of KLK7 activity, which validated the role of KLK7 as drug target for this rare disease and potentially for other common skin diseases, such as atopic dermatitis, in which KLK7 is implicated [16]. This study suggested that ABPs targeting KLKs could be exploited as theranostics given that KLK proteases could be exploited for diagnosis and/or therapy in versatile pathologies [34], including viral infections of high current interest [35]. We have also generated a phosphonate tetrapeptide ABP specific for KLK6 [36]. At the same time, another group developed tetrapeptidyl phosphonate ABPs specific for KLK2 and KLK3, and a new ABP for KLK14 all containing both natural and unnatural amino acids. Each of these ABPs carried a different fluorescent moiety in order to determine the activities of these KLKs in a multiplex assay [37] which could help deciphering the KLK activation cascade pathways (also referred to as “the KLK activome”).

#### 4.1.1. KLKs in Epidermal (Patho)Physiology

NS is a severe type of ichthyosis characterized by epidermal over-desquamation and constitutive inflammation, allergies, and a pathognomonic hair shaft defect known as bamboo hair. NS is caused by inactivating mutations in the *SPINK5* gene encoding the LEKTI inhibitor of serine proteases and is transmitted as an autosomal recessive trait [38]. *Spink5^-/-^* mice recapitulate the features of human NS and die homogeneously within 5 h from birth due to extensive dehydration caused by the severe defect of the epidermal barrier [39]. It is well-established that KLKs are expressed in the last living layer of the epidermis, named the stratum granulosum, and that KLK5 is likely the key regulator of epidermal proteolysis, since it can autoactivate and subsequently activate other KLK zymogens, namely proKLK7 and proKLK14, also present in stratum granulosum [29]. Upon zymogen activation, the activity of KLKs is quenched by binding of the LEKTI inhibitor. As the KLK-LEKTI complex diffuses towards the outermost epidermal layer, i.e., the stratum corneum, that contains anucleated keratin-filled keratinocytes (corneocytes), complex dissociation occurs because the pH is acidic in stratum corneum due to the acid mantle. This results in precise activation of KLKs at the upper layers of the stratum corneum where their activities are demanded for physiological skin renewal. Briefly, active KLKs cleave the proteins that constitute the intercellular junctions, known as corneodesmosomes, that hold together the corneocytes. In NS, LEKTI deficiency results in unopposed activities of certain KLK proteases leading to cleavage of all the stratum corneum layers, i.e., over-desquamation and associated skin barrier defect.

To demonstrate the dominant roles of KLK5 in regulating epidermal proteolysis, *Spink5^-/-^Klk5^-/-^* double knockout mice were generated, which appear normal at birth with no signs of epidermal desquamation and inflammation [40]. Nevertheless, on day 3 (P3) from birth they develop epidermal desquamation and inflammation that is gradually augmented leading the majority of mice to death by P7, while the minority of surviving mice die prematurely during the following few months [41]. Thus, solely targeting the KLK5 protease is not sufficient for long-term treatment of NS. It was demonstrated that the delayed onset of inflammation and aberrant desquamation can be rescued by eliminating *Klk7* [42] or *Tnfa* [41] on the *Spink5^-/-^Klk5^-/-^* background. The latter is especially important for treatment of NS, since anti-TNFα biologics are already approved drugs, while an anti-KLK5 agent was recently described and preclinically evaluated [43]. Indeed, when the KLK7-ABP was applied onto the epidermis of *Spink5^-/-^Klk5^-/-^* mice, it significantly attenuated the epidermal desquamation and inflammation highlighting the key role of the active KLK7 protease in inducing pathological desquamation/inflammation in this disease context [16]. The KLK7 phosphonate ABP was modified to a quenched ABP (*q*ABP) by replacing the biotin tag with the Cy5 fluorescent tag and attachment of the quencher QSY-21 to the leaving group [33] (Figure 6). This *q*ABP can be used for monitoring KLK7 activity in vivo. Overall, the combination of animal studies with ABPs enabled the validation of KLK7 as a druggable target for NS and potentially other skin pathologies that share common molecular mechanisms, such as atopic dermatitis.

#### 4.1.2. Role of KLKs in Prostate Cancer

KLK proteases have been implicated in various types of cancer [44,45,46] including prostate cancer. Prostate cancer is the second most common cancer in men second to lung cancer [47]. Increased KLK activities are thought to promote prostate tumor growth and metastasis. The reported ABPs specific for KLK2, KLK3 and KLK14 allowed their simultaneous orthogonal analysis in prostate cancer cell lines [31]. It was shown that osteoblasts secrete factors that induce the expression of KLK2 and KLK3 from LNCaP cells and that KLK14 is increased in co-cultures of osteoblasts and LNCaP but secreted only by osteoblasts. The KLK14-ABP (but not the KLK2-or KLK3-ABP) inhibited the migration of LNCaP cells engineered to express KLK14 indicating that KLK14 is required for tumor cell migration, a key step in the process of metastasis but also that ABPs can be used as inhibitors to delineate the biological role(s) of the targeted proteases [37].

#### 4.1.3. General Scheme for Synthesis of the ABPs

Peptidyl ABPs are produced in three basic steps: [1] synthesis of the tag-P4-P3-P2 tripeptide sequence by established solid phase peptide synthesis procedures, [2] synthesis of the phosphono-P1 derivative with the Oleksyszyn reaction, and [3] in solution coupling of the tripeptide with the phosphono-analogue of the P1 residue, deprotection, and purification. The synthesis scheme is outlined in Figure 7. The compounds are produced as diastereoisomers. The *R* isomer at the α-carbon is the active compound but they are almost exclusively used in racemic mixtures of the *R* and *S* isomers. Nevertheless, in the case of a new phosphonate ABP that targets matriptase different diastereomers on α-carbon were separated and tested. As expected, the compound exhibiting *R* stereochemistry over the α-carbon combined with *S* stereochemistry over the amino acid residue at the P2 position (*N*-terminal position) showed the best inhibitory properties, i.e., higher k_inact_/K_i_ [48].

### 4.2. Assays to Detect KLK Activities in Diagnostics

#### 4.2.1. ABP-Based ELISA

This assay was named ABP ratiometric (ABRA) ELISA and uses an immobilized antibody to capture the active protease-ABP adduct. The affinity tag of the ABP is used for quantification [49]. The assay was used to quantify the active KLK6 in biological and clinical specimens in order to determine the ratio of active versus total KLK6 protease. The Bio-PK probe that selectively labels trypsin-like enzymes [9] reacted with the KLK6 in the biological sample. Selectivity for KLK6 is ensured by the KLK6-capturing antibody. A schematic diagram of the ABRA-ELISA is depicted in Figure 8. Further, the ABRA ELISA was applied to determine the levels of active KLK7 using a KLK7 capturing antibody and the biotin-Phe-Phe-P(O)(OPh)_2_ KLK7-ABP [16].

#### 4.2.2. Activography

It refers to a histochemical assay developed by Pampalakis et al. [50] that can be applied to detect and spatially localize specific enzymatic activities in tissue biopsies by use of ABPs [50,51]. A schematic description of activography is provided in Figure 9. Briefly, tissue cryosections are allowed to react with a biotin-labelled ABP. Subsequently, streptavidin-horseradish peroxidase (HRP) is added and the tissue distribution of the active protease is visualized by a HRP-catalyzed chromogenic reaction. A variation of this method that used *q*ABPs had been published previously. In that case, the *q*ABP was allowed to react with the cryosections and the protease-ABP adduct was detected with fluorescence microscopy [52].

### 4.3. Cathepsins

Cathepsins constitute a group of proteases that belong to the classes of cysteine, serine, and aspartic proteases. In total, 15 cathepsins are encoded in humans [53], which play important roles in cancer, rheumatoid arthritis, atherosclerosis, and other inflammatory conditions. They are processed in the lysosomes and are active in the lysosomal acidic pH. Nevertheless, in pathological conditions, cathepsins are secreted in the extracellular milieu where they cleave components of the extracellular matrix and process cytokines and chemokines [54].

#### 4.3.1. Cathepsins in Cancer

Cathepsins have been implicated in malignant tumor growth and metastasis, since they promote tumor angiogenesis and cancer cell invasion by cleavage of the extracellular matrix. In this context, cathepsins are valuable targets for both diagnostic and therapeutic applications [54]. Especially, cathepsins B, L, and S display increased expression in cancer, and may contribute to tumor invasiveness by cleavage of E-cadherin. For this reason, a *q*ABP named YBN14 was developed that emits fluorescence when covalently bound to cathepsins and, at the same time, it can kill cancer cells by photodynamic therapy (PDT) [55]. YBN14 contains a photosensitizer that is a bacteriochlorin derivative, a selective cathepsin recognition sequence, an acyloxymethyl ketone warhead and a QC-1 quencher (Figure 10). The quencher is attached to the leaving group and is released during covalent labeling of the active cathepsin, thus, cathepsin activity is monitored by fluorescence emission. In PDT, a photosensitive molecule is irradiated (e.g., by infrared frequencies) under oxygen to generate reactive oxygen species, including singlet oxygen, that are cytotoxic.

YBN14 entails the accommodation of the *q*ABP and PDT dual platforms in a single chemical scaffold with the following advantages: (i) it works in tumor microenvironments with high cathepsin activities, (ii) it is used for real-time imaging of tumors before light sensitization, (iii) the quencher ensures diminished background fluorescence, and (iv) covalent binding prevents rapid clearance. YBN14 was evaluated in vivo, i.e., in mice bearing subcutaneous 4T1 mouse breast tumors [55]. Specifically, it targeted tumor-associated macrophages that highly overexpress cathepsins and induced apoptosis of these cells that resulted in significant tumor shrinkage. This is because tumor-associated macrophages promote cancer by chemokine and cytokine paracrine loops that induce angiogenesis, proliferation, migration, invasion, and metastasis [56]. No signs of toxicity were observed in the liver, spleen, and kidney tissues all of which express high levels of active cathepsins probably also because light treatment (illumination) is confined to the tumor site. In conclusion, YBN14 was successfully employed to detect and treat tumors in vivo.

#### 4.3.2. Atherosclerosis

Atherosclerosis involves extensive vascular inflammation that associates with deposition of cholesterol. Cysteine proteases, such as cathepsins B, L, and S, are overexpressed in vascular inflammation and are key remodeling enzymes. The YBN14 ABP was administered *iv* in *Ldlr^-/-^* mice that were fed a high fat diet. YBN14 accumulated in plaques and application of PDT resulted in reduction in lesional macrophage foam cells. This change was not accompanied by the reduction in collagen or the smooth muscle cellular content in plaques. Thus, the stability of plaques was enhanced [57].

A similar principle had been applied earlier with L-SR15 [chlorin-6e-conjugated poly(ethylene glycol)-graft-poly(D-lysine)], which is a quenched cathepsin B synthetic substrate that emits fluorescence upon cleavage by the active cathepsin. Further, the released chlorin-6e can be used for PDT. Indeed, the L-SR15 was shown to decrease macrophage infiltration in plaques in *Apoe^-/-^* mice fed a high fat diet [58]. However, substrate-based probes diffuse away from their target protease since they do not bind covalently. This is a major disadvantage of these probes compared with the ABPs that can bind by covalent bond formation and remain attached to their target. To this end, it should be noted that, as with L-SR15 for cathepsin B, ABPs for other cathepsins have been developed. In this direction, a cyanine-5 tagged ABP specific for cathepsin K was generated. Cathepsin K represents an important target for treatment of osteoporosis, since inhibition of its activity reduces the resorptive activity of osteoclasts [59].

## 5. Delineation of the Role of Neutrophil Elastase in NETosis

Neutrophils are immune cells of the innate immune system. Neutrophils contain special vesicles, also known as granules, that incorporate proteins important for microbe killing. Granules are distinguished into azurophilic granules, specific granules, gelatinase granules, and secretory vesicles. Azurophil granules contain the following four serine proteases: neutrophil elastase (NE), cathepsin G, proteinase 3 (PR3), and neutrophil proteinase 4 (NSP4), also known as PRSS57 [60]. Neutrophils prevent spreading of pathogens by formation of neutrophil extracellular traps (NETs) composed of DNA extruded from the cells in a process known as NETosis [61]. It was hypothesized that NE or the activities of the other azurophil proteases participate in the process of NETosis via proteolysis of nuclear proteins to induce chromatin decondensation. To identify if the NE activity is indeed required for NETosis, specific ABPs that target these enzymes were generated [62] and used to block the azurophil proteases in an effort to unravel their role in NETosis [63]. These ABPs are *O,O*-diphenyl phosphonate tetrapeptides with an *N*-terminal biotin or a fluorescent moiety. Exploitation of these ABP inhibitors in functional assays revealed that the catalytic activities of azurophil proteases are not required for the formation of NETs. This is another characteristic example of how ABP-inhibitors can be used in molecular cell biology studies to decipher the biological functions of the target enzymes [63]. Another ABP that was also developed for NE carries a sulfonyloxyphthalimide warhead [64]. Finally, a peptide multi-branched “supersilent” probe was generated to monitor the activity of NE, which emits a huge fluorescence signal upon cleavage by the NE, based on which activated macrophages expressing elevated NE activity can be detected and monitored [65].

## 6. From Contrast Agents to Radiopharmaceuticals

X-ray computed tomography (CT) is an imaging method with excellent spatial resolution. A new iodinated nanoscale ABP that binds to cathepsins B, L, and S was developed as contrast agent based on iodine that is primarily used in CT. It carries a short cathepsin-targeting peptide sequence, an acyloxymethyl ketone warhead, and a dendrimer tag that can accommodate up to 48 iodine atoms [66], as shown in Figure 11. This ABP provided detectable CT contrast when it was administered in mice bearing subcutaneous 4T1 breast tumors.

Substitution of iodine atoms with radioactive iodine will yield an ABP-based radiopharmaceutical for combined detection of tumors using radiology-based medical imaging and for anticancer therapy by targeting radioactivity selectively to tumor cells.

## 7. Conclusions-Future Perspectives

ABPs are novel interesting molecules that enable the selective detection of active proteases in complex biological settings. Screening of chemical libraries of synthetic peptides that consist of natural and/or unnatural amino acids, and other methodologies, allow determination of the substrate specificity of proteases. Specific substrate recognition sequences can be exploited to the design and synthesis of new peptidyl phosphonate-ABPs that act as suicide inhibitors that bind covalently to the active site of the specific protease.

Consequently, these specific ABPs not only label the target protease but also inhibit its activity and can be used as candidate pharmaceutical compounds. Indeed, the successful application of ABPs to treat the severe ichthyosis NS, inhibit the migration of prostate cancer cells, treat breast cancer and atherosclerosis has been demonstrated. Cumulatively, ABPs emerge as new theranostic agents, namely compounds that accommodate a detection and a therapeutic moiety in a single chemical scaffold. The term “theranostic” is usually used in nanotechnology to describe nanoparticles with dual diagnostic and therapeutic potential [67]. ABPs expand the field of theranostics to encompass small synthetic molecules. Up to date, application of ABPs has largely focused on their use in the profiling of active proteases (or enzymes, in general), a process known as ABP profiling and not as inhibitors to elucidate the complex biological functions of proteases/enzymes or to treat a given disease in which the protease/enzyme is involved. In this direction, we outline successful applications of ABPs as inhibitors of proteases with the aim to pinpoint their potential as putative therapeutics and as tools to unravel the functional roles of the target proteases.

Within recent years, the field of ABPs has expanded and a large chemical variety of ABPs are now present that enable the targeting of a large variety of proteases. It is suggested that these ABPs can be directly exploited for the functional characterization of a protease in a given disease. In this way, we will increase the number and availability of new compounds with potential pharmacological action. One major issue with drugs is the fact that they have off-target effects. These may sometimes increase their potency or more often account of adverse effects. The drugs can be designed to carry a detection tag and, in this way, to generate novel ABPs. Since ABPs not only inhibit but also label their biological targets, they can be exploited in therapeutics to identify all potential off-target effects in an ABP profiling study. This knowledge will further assist in the design of better therapeutic agents deprived from severe adverse effects.

## Figures and Tables

**Figure 1 pharmaceutics-14-00977-f001:**
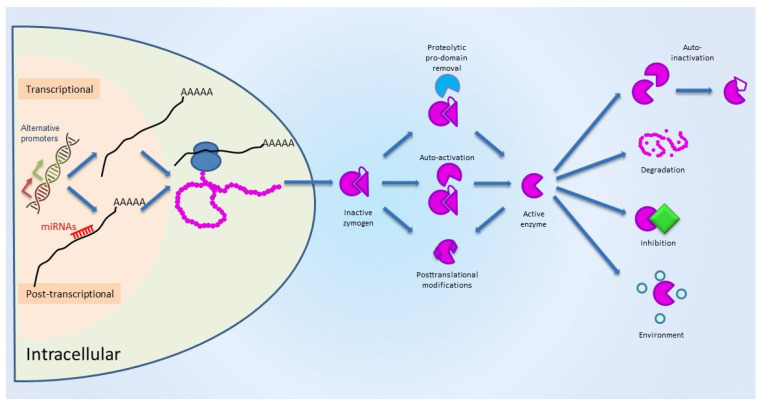
Proteases are controlled at multiple levels. Proteases are regulated at the level of transcription through alternative promoters, production of splice variants, and DNA/promoter methylation, and post-transcriptionally through miRNAs and alternative polyadenylation signals. Fine tuning of the enzymatic activities of proteases is additionally achieved post-translationally, since they are synthesized as zymogens that require activation. In turn, active proteases may be inhibited via binding to specific endogenous inhibitors, while their activities can be altered by changes in the pH, by chemical modifications or cleavage by other proteases or degradation (e.g., by the ubiquitin-proteasome system).

**Figure 2 pharmaceutics-14-00977-f002:**
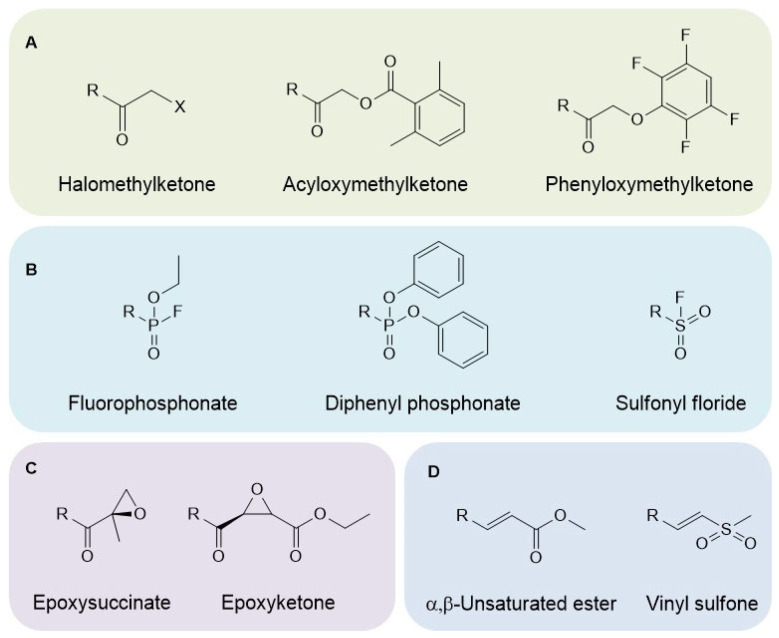
Chemical formulas of warheads used to target proteases. Phosphonates are the classical warheads for targeting serine proteases and acyloxymethyl ketones for cysteine proteases. (**A**), activated ketones. (**B**), phosphonates. (**C**), epoxides. (**D**), Michael acceptors.

**Figure 3 pharmaceutics-14-00977-f003:**
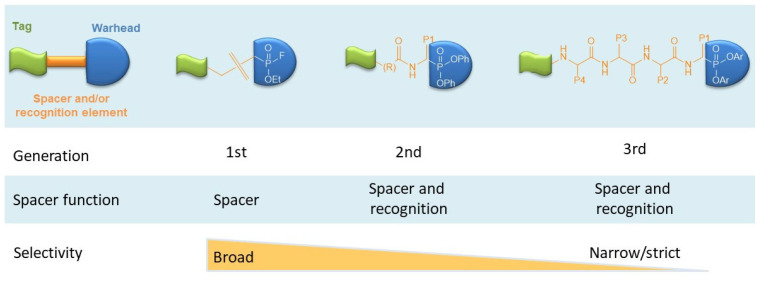
Transition from first to third generation phosphonate ABPs. Incorporation of a longer recognition element in the ABP molecule results in improved specificity. Notably, first generation phosphonate ABPs target hydrolases including proteases.

**Figure 5 pharmaceutics-14-00977-f005:**
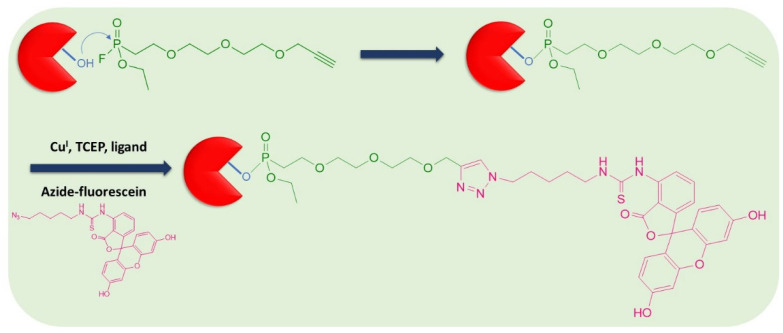
Click chemistry-based labeling of serine hydrolases. The active enzyme is allowed to react with the 1-alkyne phosphonate derivative. Then, the tag, in this case an azido derivative of fluorescein, is added to form the final adduct.

**Figure 6 pharmaceutics-14-00977-f006:**
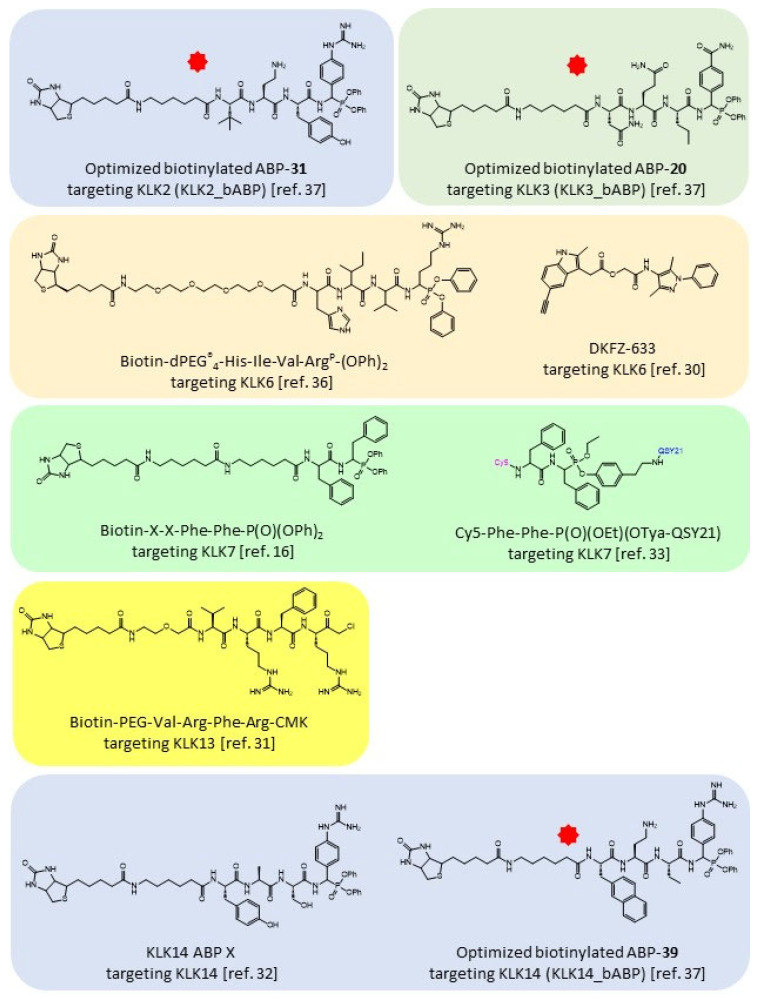
Chemical structures of ABPs targeting active KLKs. The chemical formulas of all reported ABPs that bind to KLKs are shown. Red stars indicate that analogues of these molecules labelled with fluorescent moieties have been generated.

**Figure 7 pharmaceutics-14-00977-f007:**
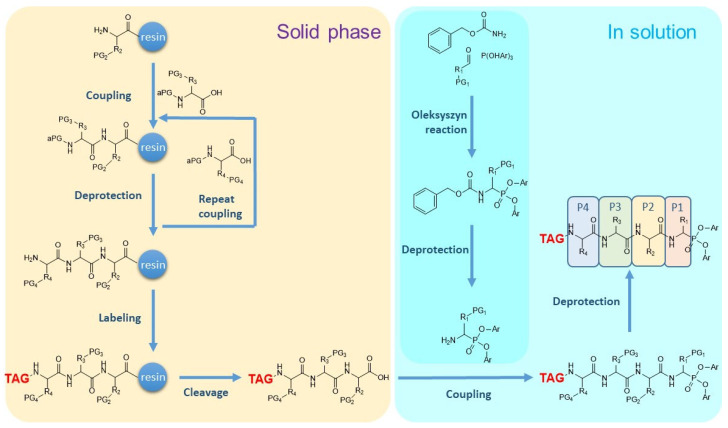
Schematic diagram of the synthesis of the peptidyl-based phosphonate ABPs. Briefly, synthesis is carried out in three steps. First the tagged and protected peptide encompassing the P4-P3-P2 amino acid residues is synthesized by standard solid phase peptide synthesis. Then, the phosphono-P1 residue is synthesized, coupled to the tagged tripeptide, and the product is deprotected and purified.

**Figure 8 pharmaceutics-14-00977-f008:**
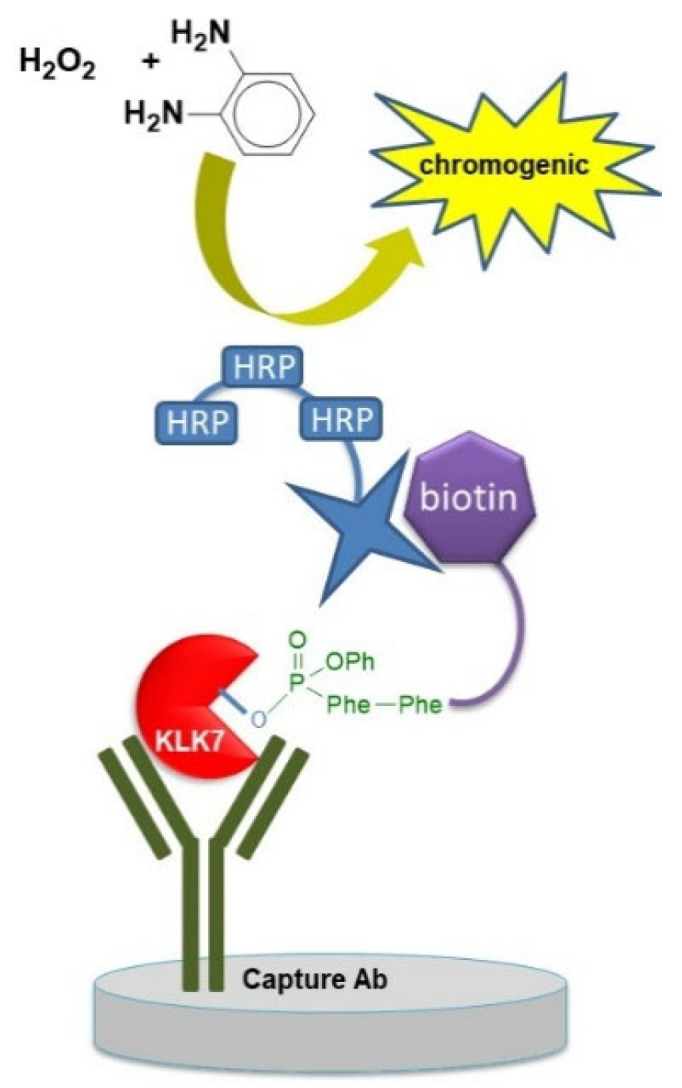
Schematic representation of the ELISA used to determine active KLKs. The ABP is allowed to react with the target protease in the biological/clinical sample. Then, the sample is applied onto a multi-well plate covered with the capturing antibody. For detection, a streptavidin-based system is added (here streptavidin depicted as star with an HRP-polymer is shown). Detection may be carried out with a chromogenic reaction (as shown) or by monitoring the time-resolved fluorescence emission.

**Figure 9 pharmaceutics-14-00977-f009:**
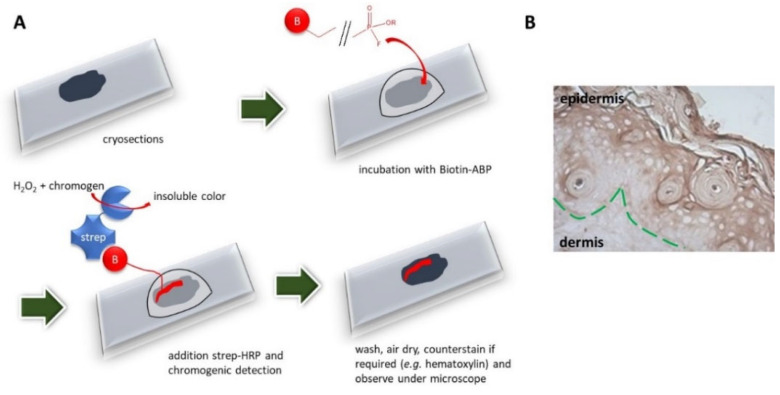
Activography. (**A**) Schematic diagram of activography. Tissue cryosections are incubated with a biotin-labelled ABP, then, streptavidin-HRP is added and the protease-ABP adducts are detected by an HRP-catalyzed chromogenic reaction. (**B**) Example of activography conducted on the skin biopsy obtained on P7 from *Spink5^-/-^Klk5^-/-^* mice with the B42P probe showing increased proteolytic activities in the epidermis and stratum corneum.

**Figure 10 pharmaceutics-14-00977-f010:**
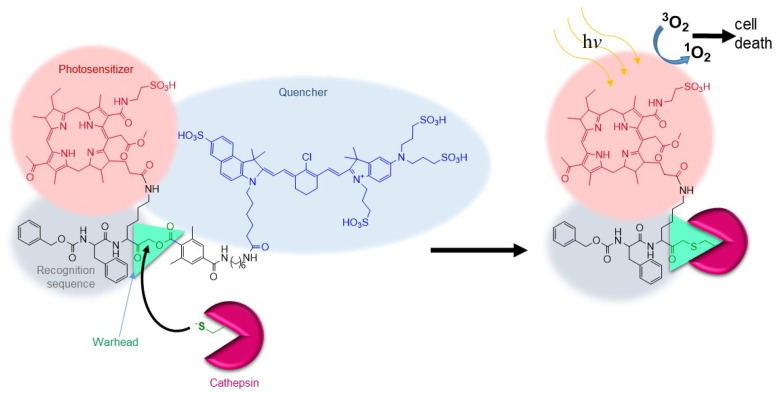
Chemical structure of YBN14 and labeling activity.

**Figure 11 pharmaceutics-14-00977-f011:**
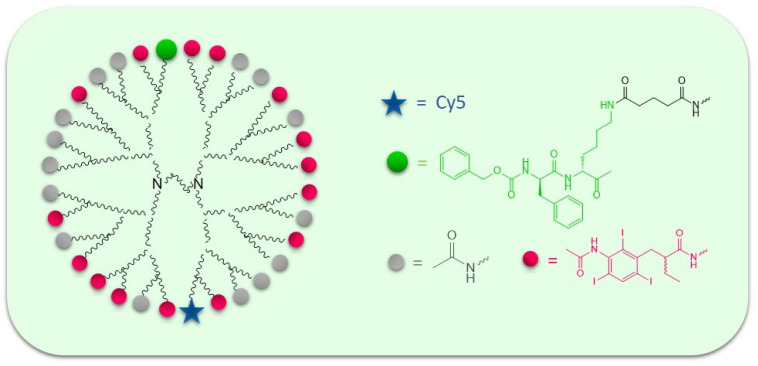
Structure of the iodinated nanoscale ABP. The ABP comprises the recognition sequence carrying an acyloxymethyl ketone. The detection unit is based on a polyamidoamine (PAMAM) core dendrimer on which 16 iodine tags are attached, each containing three iodine atoms.

## Data Availability

Not applicable.

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
