# Peer review of "Activity-Based Probes for Proteases Pave the Way to Theranostic Applications"

_pharmaceutics, 2022, doi:10.3390/pharmaceutics14050977_

Round 1
Reviewer 1 Report
The authors provide an instructive, accurately written review of very high quality. It should be enlarged somehow to become more comprehensive.
In the first paragraph of chapter 2, serine and cysteine proteases are mentioned to be addressed by ABPs. However, there are few examples that also non-covalently acting protease can be labeled by such probes. I recommend the exemplary notation of cyanine-3 containing ABPs for matrix metalloprotaeses (Kaminska et al. Angew. Chem. Int. Ed. 2021, 60, 18272).
In chapter 2, one recent review (2020, ref 9) on ABPs for proteases has been included. I suggest also to cite the following (Rodriguez-Rios et al. Chem. Soc. Rev. 2022, 51, 2081).
ELISA and activography are nicely specified in chapters 4.1.1. and 4.2.2. In chapter 2, the other opportunities of analytical utilizations of ABPs for proteases should be noted, e.g. Western blotting, confocal microscopy, proteome analysis and more. An example for protease profiling with APBs for SUMO-specific proteases was reported by the group of the late Huib Ovaa, which should be noticed (Mulder et al. Angew. Chem. Int. Ed. 2018, 57, 8958). The combination of size-exclusion chromatography and HPLC fluorescence detection was first demonstrated with a phosphonate ABP, which should be mentioned (Häußler et al. Chem. Eur. J. 2017, 23, 5205).
In Figure 4, positional scanning of synthetic combinatorial libraries is noted. A recent, important example should mentioned showing the application of this technique for the development of cyanine-5- or Bodipy-based ABPs with a vinylsulfone warhead for the main protease (Mpro) of SARS-CoV-2 (Rut at. al. Nat. Chem. Biol. 2021, 17, 222).
Page 6, the abbreviation NS has to be explained, i.e. Netherton syndrome.
It is noted (page 8 and Figure 7) that the peptidyl phosphonates were applied as racemic mixtures. However, it has been demonstrated that such a probe with a defined stereochemistry, after separation of the single epimer, was employed to label the serine protease matriptase-2. This study should be mentioned (Häußler et al. Chem. Eur. J. 2016, 22, 8525) and added to the list of references.
In chapter 4.3.1 (cathepsins), a recent study on cyanine-5 containing probes with extraordinary potency for cathepsin K should be included (Lemke et al. J. Med. Chem. 2021, 64, 13793).
Concerning chapter 5 (neutrophil elastase), two publications should be taken into account. A sensitive optical probe for the detection of activated neutrophils and NETs was developed and shown to be specific for human neutrophil elastase (Rios et al. Chem. Commun. 2021, 57, 97). A Lossen rearrangement was implemented for a double FRET by which human neutrophil elastase can be analyzed with a highly potent probe with a sulfonyloxyphthalimide warhead (Schulz-Fincke et al. Biochemistry 2018, 57, 742).
Some line breaks, probably not used by the authors, are inapt, e.g. proteol-ysis (page 1), tran-scription (page 2 and 5), strep-tavidin (page 10).
Correct "a,b-Unsaturated" (Figure 2) by using Greek letters. Correct to "azurophilic" (page 12).
Author Response
Reviewer 1
The authors provide an instructive, accurately written review of very high quality. It should be enlarged somehow to become more comprehensive.
In the first paragraph of chapter 2, serine and cysteine proteases are mentioned to be addressed by ABPs. However, there are few examples that also non-covalently acting protease can be labeled by such probes. I recommend the exemplary notation of cyanine-3 containing ABPs for matrix metalloprotaeses (Kaminska et al. Angew. Chem. Int. Ed. 2021, 60, 18272).
Answer:
The suggested reference has been added and discussed in chapter 2 as follows “It must be noted that ABPs have been developed for the metalloproteinase 12 (MMP12) that lacks a canonical nucleophile in its active site but contains Zn2+ instead. To design the ABP, a known competitive inhibitor of MMP12 was modified to encompass a cleavage linker conjugated to a fluorescent tag (i.e., cyanine 3). Once the ABP is bound onto the active site, cleavage of the linker occurs, and the released cyanine 3-carrying group covalently modifies neighboring nucleophiles (side chains of various residues that face towards the active site), a process known as proximity driven reaction [5].”
In chapter 2, one recent review (2020, ref 9) on ABPs for proteases has been included. I suggest also to cite the following (Rodriguez-Rios et al. Chem. Soc. Rev. 2022, 51, 2081).
Answer:
This interesting recent review article has been added discussed in chapter 2 as follows “Finally, there are probes for proteases known as substrate probes. These molecules are fluorescently quenched peptide substrates encompassing the protease substrate recognition sequence. Cleavage of the peptide by the active protease releases the quencher and the proteolytic activity is quantified by monitoring fluorescence emission [6].”
ELISA and activography are nicely specified in chapters 4.1.1. and 4.2.2. In chapter 2, the other opportunities of analytical utilizations of ABPs for proteases should be noted, e.g. Western blotting, confocal microscopy, proteome analysis and more. An example for protease profiling with APBs for SUMO-specific proteases was reported by the group of the late Huib Ovaa, which should be noticed (Mulder et al. Angew. Chem. Int. Ed. 2018, 57, 8958). The combination of size-exclusion chromatography and HPLC fluorescence detection was first demonstrated with a phosphonate ABP, which should be mentioned (Häußler et al. Chem. Eur. J. 2017, 23, 5205).
Answer:
The suggested references have been added and discussed in page 4 as follows “ABPs for proteases can be applied for analytic purposes in Western blotting, SDS-PAGE coupled to fluorescence imaging, confocal laser scanning microscopy, and activity-based proteomic profiling as, for example, described for ABPs targeting SUMO proteases [12]. The ABP-enzyme adduct can be detected by monitoring fluorescence emission following size exclusion chromatographic separation of the reaction products between the ABP and the protease, as demonstrated for the phosphonate ABP specific for matriptase [13].”
In Figure 4, positional scanning of synthetic combinatorial libraries is noted. A recent, important example should mentioned showing the application of this technique for the development of cyanine-5- or Bodipy-based ABPs with a vinylsulfone warhead for the main protease (Mpro) of SARS-CoV-2 (Rut at. al. Nat. Chem. Biol. 2021, 17, 222).
Answer:
The suggested reference has been added and discussed on page 4 as follows “Here, it should be mentioned that positional scanning libraries containing both natural and unnatural amino acids have been used to develop an ABP for the main protease (Mpro) of SARS-CoV-2 that is one of the most important targets for the development of antiviral drugs. Initially, screening of the library identified the best substrate that was converted to an ABP by addition of biotin or fluorescent (cyanine-5 or bodipy) tags at the N-terminus and a vinyl sulfone warhead at the C-terminus [11].”
Page 6, the abbreviation NS has to be explained, i.e. Netherton syndrome.
Answer:
It has been explained in line 165 when first introduced.
It is noted (page 8 and Figure 7) that the peptidyl phosphonates were applied as racemic mixtures. However, it has been demonstrated that such a probe with a defined stereochemistry, after separation of the single epimer, was employed to label the serine protease matriptase-2. This study should be mentioned (Häußler et al. Chem. Eur. J. 2016, 22, 8525) and added to the list of references.
Answer:
The suggested reference has been added and discussed on page 9 as follows “Nevertheless, in the case of a new phosphonate ABP that targets matriptase different diastereomers on α-carbon were separated and tested. As expected, the compound exhibiting R stereochemistry over the α-carbon combined with S stereochemistry over the amino acid residue at the P2 position (N-terminal position) showed the best inhibitory properties, i.e., higher kinact/Ki [48].”
In chapter 4.3.1 (cathepsins), a recent study on cyanine-5 containing probes with extraordinary potency for cathepsin K should be included (Lemke et al. J. Med. Chem. 2021, 64, 13793).
Answer:
The suggested reference has been added and discussed on page 13 as follows “To this end, it should be noted that as with L-SR15 for cathepsin B, ABPs for other cathepsins have been developed. In this direction, a cyanine-5 tagged ABP specific for cathepsin K was generated. Cathepsin K represents an important target for treatment of osteoporosis, since inhibition of its activity reduces the resorptive activity of osteoclasts [59].”
Concerning chapter 5 (neutrophil elastase), two publications should be taken into account. A sensitive optical probe for the detection of activated neutrophils and NETs was developed and shown to be specific for human neutrophil elastase (Rios et al. Chem. Commun. 2021, 57, 97). A Lossen rearrangement was implemented for a double FRET by which human neutrophil elastase can be analyzed with a highly potent probe with a sulfonyloxyphthalimide warhead (Schulz-Fincke et al. Biochemistry 2018, 57, 742).
Answer:
The suggested references have been added and discussed on page 13 as follows “Another ABP that was also developed for NE carries a sulfonyloxyphthalimide warhead [64]. Finally, a peptide multi-branched “supersilent” probe was generated to monitor the activity of NE, which emits a huge fluorescence signal upon cleavage by the NE, based on which activated macrophages expressing elevated NE activity can be detected and monitored [65].”
Some line breaks, probably not used by the authors, are inapt, e.g. proteol-ysis (page 1), tran-scription (page 2 and 5), strep-tavidin (page 10).
Answer:
They have been added automatically by the parameters set in the template.
Correct "a,b-Unsaturated" (Figure 2) by using Greek letters. Correct to "azurophilic" (page 12).
Answer:
They have been corrected.

Reviewer 2 Report
The manuscript entitled „Activity-based probes for proteases pave the way to theranostic applications” has attention to the possibility of using proteases in the construction of activity-based probes.
The manuscript is well-organised, but for the readers' convenience, it is commendable to address the following issues:
- a short introduction before Section 1, would help the readers to better understand the purpose of the present review. The authors can consider addressing the multiple possibilities for using the proteases.
- Carefully check all the figures from the manuscript. In most of them, the text is written in a hardly visible text.
- Where is the case; mention if the reprint of the figures is permitted.
- Consider a sub-section that classifies the proteases
- Conclusions should highlight the main advantages of using ABP for proteases.
- Add a section/sub-section that refers to future perspectives.
Author Response
Reviewer 2
The manuscript entitled “Activity-based probes for proteases pave the way to theranostic applications” has attention to the possibility of using proteases in the construction of activity-based probes.
The manuscript is well-organised, but for the readers' convenience, it is commendable to address the following issues:
- a short introduction before Section 1, would help the readers to better understand the purpose of the present review. The authors can consider addressing the multiple possibilities for using the proteases.
Answer:
The focus of the review is outlined just above section 1 as follows “In this sense, ABPs represent new theranostic agents. We outline recent developments pertaining to ABPs for proteases with potential therapeutic applications, with the aim to highlight their importance in theranostics.” To avoid redundancy, it is not repeated in Section 1 just after the abstract
In section 1 it is written that (proteases) “They are involved in almost all biological processes in an organism.”
Finally, we have selected to present proteases, because it is the only class of enzymes where the theranostic application of ABP has been demonstrated.
- Carefully check all the figures from the manuscript. In most of them, the text is written in a hardly visible text.\
Answer:
All figures have been checked revised accordingly.
- Where is the case; mention if the reprint of the figures is permitted.
Answer:
All figures were created by our group and thus no copyright issues are involved.
- Consider a sub-section that classifies the proteases
Answer:
The classification of proteases is a textbook material but it is summarized in the section 1 in first paragraph “Proteases are enzymes that catalyze the hydrolysis of peptide bonds. They are involved in almost all biological processes in an organism. The term “degradome” was introduced to describe “the complete set of proteases that are expressed at a specific time by a cell, tissue or organism” [1]. The human genome encodes 553 proteases that are classified based on the catalytic mechanism into the following five categories: aspartic, metallo, cysteine, serine, and threonine proteases. A wide spectrum of pathologies is characterized or triggered by abnormal expression and/or activation of proteases and/or endogenous inhibitors of proteases, thus, tight regulation of proteolysis is indispensable for maintenance of homeostasis in biological systems [2]. ”
- Conclusions should highlight the main advantages of using ABP for proteases.
Answer:
In the conclusions section, we emphasize the use of ABP for theranostic applications. ABPs exist for several years but they have only been applied as diagnostics.
Nevertheless, this important point is demonstrated in detailed throughout chapter 4, where the application of the ABPs in several diseases is given.
- Add a section/sub-section that refers to future perspectives.
Answer:
The section 6 is a representative of potential future perspectives.

Reviewer 3 Report
The review by Sotiropoulou describes the use of activity based probes to characterise proteases. The manuscript is well written and contains many informative figures. I noted only two problems. In Figure 7 the structure shown after the coupling step needs to have "R1" changed to "R2" and "R2" to "R3". Line 294 seems to be missing a word or words between "are" and "by".
Author Response
Reviewer 3
The review by Sotiropoulou describes the use of activity based probes to characterise proteases. The manuscript is well written and contains many informative figures. I noted only two problems. In Figure 7 the structure shown after the coupling step needs to have "R1" changed to "R2" and "R2" to "R3".
Answer:
The Figure has been revised accordingly.
Line 294 seems to be missing a word or words between "are" and "by".
Answer:
The word detected has been added between are and by.

Round 2
Reviewer 2 Report
Most of the reviewer's comments are not satisfactorily addressed.
Moreover, the authors should consider that as long as in the figures' legends appear references, then it should be mentioned that the figure belongs to the authors and does not have anything in common with the mentioned work.
Conclusions should highlight the main findings even if it is already described in section 4.
Future perspectives - section 6 might be representative for the recommended section but is not sufficiently highlighted as future perspectives.
Author Response
Moreover, the authors should consider that as long as in the figures' legends appear references, then it should be mentioned that the figure belongs to the authors and does not have anything in common with the mentioned work.
Answer:
We have already responded to this comment that was made during R1 by the Assistant Editor as follows: “we would like to mention that this figure was created by our group and the references cited in the figure legend are only to indicate where the reader should look for detailed information, thus, no copyright permission is required.”
If the figure had been obtained from another source, we would have written “adapted from…” which is not the case.
Conclusions should highlight the main findings even if it is already described in section 4.
Answer:
We have now added in the conclusions the following: “Indeed, the successful application of ABPs to treat the severe ichthyosis NS, inhibit the migration of prostate cancer cells, treat breast cancer and atherosclerosis has been demonstrated.”
Future perspectives - section 6 might be representative for the recommended section but is not sufficiently highlighted as future perspectives.
Answer:
We have added the following in the Conclusions-Future perspectives section: “Within the last years, the field of ABPs has expanded and a large chemical variety of ABPs are now present that enable the targeting of a large variety of proteases. It is suggested that these ABPs can be directly exploited for the functional characterization of a protease in a given disease. In this way, we will increase the number and availability of new compounds with potential pharmacological action. One major issue with drugs, is the fact that they have off-target effects. These may sometimes increase their potency or more often account of adverse effects. The drugs can be designed to carry a detection tag and, in this way, to generate novel ABPs. Since ABPs not only inhibit but also label their biological targets, they can be exploited in therapeutics to identify all potential off-target effects in an ABP profiling study. This knowledge will further assist in the design of better therapeutic agents deprived from severe adverse effects.”
